# Spotting Green Tides over Brittany from Space: Three Decades of Monitoring with Landsat Imagery

Louise Schreyers [1,*], Tim van Emmerik [1], Lauren Biermann [2] and Yves-François Le Lay [3]

1   Hydrology and Quantitative Water Management Group, Wageningen University,
    6708 PB Wageningen, The Netherlands; tim.vanemmerik@wur.nl
2   Plymouth Marine Laboratory, Plymouth PL1 3DH, UK; lbi@pml.ac.uk
3   École Normale Supérieure de Lyon, 69007 Lyon, France; yves-francois.le-lay@ens-lyon.fr
*   Correspondence: louise.schreyers@wur.nl

**Abstract:** Green tides of macroalgae have been negatively affecting the coasts of Brittany, France, for at least five decades, caused by excessive nitrogen inputs from the farming sector. Regular areal estimates of green tide surfaces are publicly available but only from 2002 onwards. Using free and openly accessible Landsat satellite imagery archives over 35 years (1984–2019), this study explores the potential of remote sensing for detection and long-term monitoring of green macroalgae blooms. By using a Google Earth Engine (GEE) script, we were able to detect and quantify green tide surfaces using the Normalized Difference Vegetation Index (NDVI) and Normalized Difference Water Index (NDWI) at four highly affected beaches in Northern Brittany. Mean green tide coverage was derived and analyzed from 1984 to 2019, at both monthly and annual scales. Our results show important interannual and seasonal fluctuations in estimated macroalgae cover. In terms of trends over time, green tide events did not show a decrease in extent at three out of four studied sites. The observed decrease in nitrogen concentrations for the rivers draining the study sites has not resulted in a reduction of green tide extents.

**Keywords:** earth observation; macroalgae; Google Earth Engine; remote sensing; eutrophication; satellite; coastal waters

## 1. Introduction

Macroalgae blooms in coastal waters are common in several regions of the world. As many as 43 regions are regularly affected by green tide events worldwide, of which 19 are located in Asia, 11 in Europe, and 9 in North-America [1]. The world's largest green tide took place along the Yellow Sea coast during the Olympic Games sailing competition of 2008, in the Qingdao region of China. Emergency response costs were estimated at 200 million euros and losses from the aquaculture sector amounted to 86 million euros [1]. Since 2011, an intensification in brown macroalgae blooms has been reported in the Caribbean, Gulf of Mexico, and West Africa, with recurring events since then [2–5]. These large stranding of brown *Sargassum* algae bear similarities with green tide events, in terms of recurrence, extent, seasonality, and the likelihood of excessive nutrient inputs being a driver [5]. Macroalgae blooms result from an excessive growth of plant biomass, when inputs of nitrogen or phosphorus, carried by rivers into the sea, exceed the natural assimilative capacity of coastal ecosystems [6–9]. The currents and tides uproot these algae, leaving them stranded on the beaches, creating the commonly called green tides. In addition to the nutrient flow, the proliferation of macroalgae also requires specific coastal conditions, such as confinement of water masses, broad and flat shores, shallow depths, low turbidity levels, and adequate lightening, that enable a rapid warming of the water column [10–12].

Green macroalgae proliferations are a persistent issue in Brittany, France. Reports on green tides events were made since as early as the 1970s [12,13]. Brittany is considered the

worst affected region in France in terms of intensity and frequency, with green tides from the genus *Ulva* occurring every year [11]. The green tides occurring in Brittany became a national issue with headline news in 2008 of the death of a horse and loss of consciousness of its rider after they breathed the toxic seaweed fumes (hydrogen sulfide) [14]. The strong suspicions from environmental associations and health professionals that the hydrogen sulfide emanating from the washed algae's decomposition might pose serious threats to human and animal health, including death when in high concentrations, led to the creation of the first Anti-Algae Plan in 2010 by the French government [15]. Along with these reduction efforts, regular monitoring of the green tide surfaces have been undertaken by the French Algae Technology and Innovation Center (CEVA). CEVA has been using aerial photography to quantify the algae surface on 95 sites. However, CEVA does not publicly detail its methodology and the surface estimates were not systematically made prior to 2002. A replicable and low-cost approach to estimate green tides extent would be beneficial considering the increase in macroalgae blooms worldwide and the need for additional monitoring efforts.

Earth Observation (EO) has been used extensively to detect algae blooms and aquatic vegetation in both marine and coastal ecosystems [16–21]. Its main advantage lies in the capacity of covering large geographical areas and enabling time series analysis, allowing both near real-time and long-term monitoring. Following the large-scale outbreak of green macroalgae bloom in China in 2008, numerous studies used remote sensing to better understand this massive spread [22–25]. Overall, EO has proved effective in characterizing various aspects of algae blooms, such as their origin, distribution patterns, occurrence and frequency, severity of events, surface and biomass estimates, temporal changes, and potential causes. Long-term analysis remained scarce, however, due to data accessibility costs, constraints, and long processing times.

New opportunities have arisen with long-term archives being made publicly available and the use of cloud platform for fast processing of hundreds of images at a time. Since 2008, the Landsat program's entire data archive (from 1974) and its new products have been made available to the general public free of charge [26]. This has generated a new paradigm in the field of EO, enabling long-term monitoring of ecosystems [26]. However, multi-temporal analysis has remained considerably constrained by computing capabilities until recently, since the manual downloading, preprocessing, analyzing and storing of satellite imagery is both time-consuming and power-intensive. The recent emergence of cloud processing platforms, such as the Google Earth Engine (GEE), which gathers multi-petabytes of geospatial datasets and satellite imagery, including all Landsat and Sentinel-2 archives, overcomes this constraint. To date, GEE is the largest free-to-use EO data library [27]. It enables the fast processing of large amount of data directly online, using Google's computational infrastructure [27]. Global scale assessment of ecosystems over decades were made possible with the GEE, for instance, for beaches [28] or water surface changes [29].

In this paper, we aim to: (1) demonstrate the detectability of stranded green macroalgae on the foreshores using Landsat imagery; and (2) estimate green tide surfaces over 35 years (1984–2019) at four affected beaches in Northern Brittany. Time-series analysis provides insights on the annual trends of green tide extents, as well as on their seasonal cycle of appearance and decline. Reconstructing such a long time-series has two main interests. Firstly, it provides a baseline to assess the efficiency of the French governmental measures to combat green tides. Secondly, the simple method built to estimate algae deposition on beaches with open-access imagery and in a fast processing environment can be reused in other regions around the world also affected by algae blooms.

## 2. Materials and Methods

### 2.1. Site Selection

Four different sites were selected to monitor green tides: Saint-Michel-en-Grève, Locquirec, Yffiniac, and Morieux. The first two are located a few kilometers west of the

town of Lannion, the two others in the bay of Saint-Brieuc (Figure 1). These two sectors are distant from each other by approximately 76 km. The four sites are all located in the Côtes d'Armor *département* in Northern Brittany. Blooms in this part of Brittany are essentially composed of the *Ulva armoricana* species, although other *Ulva* species have been documented in the rest of Brittany [11,12,30]. *Ulva* are benthic algae species, and as such are fixated at the bottom of the water in coastal areas. Due to the mechanical action of the waves, the macroalgae can get torn from the bottom of the water and be resuspended at the water surface. They eventually end up stranded on beaches. *Ulva* macroalgae are opportunistic species, able to combine rapid growth in favorable conditions (in spring, when illumination increases) with high survival rates during the winter months. In the fall and winter months, the green macroalgae are able to survive at the bottom of the water without notable growth nor regression of their biomass [12].

These characteristics, along with favorable site conditions for algae growth and important influx of nutrients, explain that certain sites are impacted by green tide events each year. This is the case of the four studied sites, with stranding events recorded every year between 1997 and 2018 [31]. Given that our approach focuses on multi-temporal analysis, it was crucial to select sites that had frequent green tides and not merely sporadic events.

Due to the high frequency of green tide events and their amplitude, national public efforts have been deployed in the past 10 years to fight against the algae proliferation in these sites, allowing us to assess the efficiency of these measures. Both the Bay de Lannion area and the bay of Saint-Brieuc are among the eight bays targeted by the first Anti-Algae Plan launched in 2010 [32], as well as its ongoing second iteration, from 2017 to 2021 [33]. For the first plan, 134 million of euros were spent, and another 50 million were budgeted for the second plan [12,33]. Lastly, CEVA has been monitoring the green tide events closely since 2002, allowing us to compare our surface estimates with their trends.

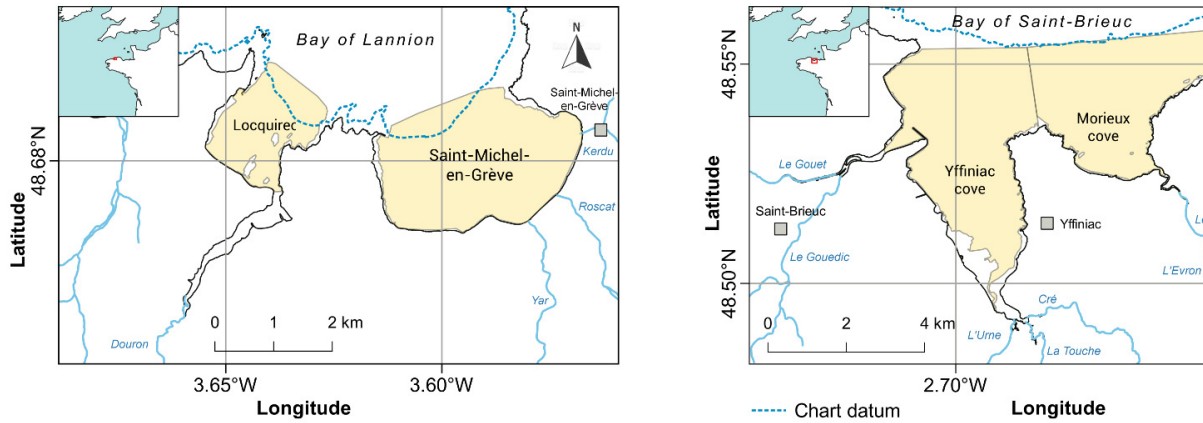

**Figure 1.** Localization of the four studied sites. Note the different scales used for the two bays. Total beach areas are of approximately 1246, 878, 429, and 140 hectares (respectively: Yffiniac, Morieux, Saint-Michel-en-Grève, and Locquirec).

## 2.2. Imagery Selection

In this study, we used Google Earth Engine (GEE) as the main satellite imagery processing tool for detecting macroalgae tides on the shores of the study sites. A script was created in GEE to filter Landsat imagery by date, location, cloud coverage and to import the geometries of the studied sites. The scripts are openly available. Multi-temporal analysis requires removing atmospheric effects. That is why surface reflectance (SR) Landsat products were preferred to Top-of-Atmosphere (TOA) scenes.

We relied on the Landsat 5 Thematic Mapper (TM) and Landsat 8 Operational Land Imager (OLI) datasets, which together enable an almost yearly continuous coverage of the time period ranging from 1984 to 2019. Landsat 5 and 8 data were obtained for the seven months of April, May, June, July, August, September, and October. This choice was based on existing literature which highlights that the stranding of *Ulva* macroalgae can

start by mid-April, and the accumulation on beaches begins to recede from September to October [11,12,30–32]. It should be noted that, when on the foreshore, the macroalgae are not a living organism anymore [12]. Thus, the stranding season might differ slightly from the growing season of the living green macroalgae, in the sense that a delay can occur between the growth peak of macroalgae and the days registering the largest green tides on the foreshore of beaches.

The months of May, June, July, August, and September correspond to the period of maximum extent of green algae in Northern Brittany, thus being retained for computing annual and 5-year mean in green tides.

For the four sites, Landsat SR scenes were selected in considerations of cloud conditions. Scenes with cloud coverage between 0 and 20% were retained for analysis. This considerably limited the overall number of scenes ($n = 192$, $n = 189$, $n = 146$, and $n = 128$ for Saint-Michel, Locquirec, Yffiniac, and Morieux, respectively), given the frequent cloud coverage over Northern Brittany. Secondly, only scenes corresponding to low tides were selected. Since this study focuses on the green algae stranded on the intertidal zone, low tide conditions were necessary for surface estimations. The selection of scenes during low tide conditions was done through visual inspection. A total of 102 scenes were retained for analysis for Saint-Michel, 101 for Locquirec, 90 for Yffiniac, and 68 for Morieux. Annual green macroalgae surface estimates were calculated by averaging the surface estimates for all images during the months of May, June, July, August, and September. The images for the months of April and October were only used for characterizing the seasonality of green tides.

Figure 2 details the number of scenes that were then analyzed to estimate the algae surface for each site, considering the 5-month and the 7-month periods. Yearly variation in the number of scenes arise mainly due to variations in cloud coverage and the availability of images in low tide conditions. Note that, for some years, no adequate imagery has been found. This could be due to the absence of clear conditions or, in the case of the year 2012, to the down-scaling of the Landsat 5 mission before Landsat 8 became operational. The use of Landsat 7 was considered to fill in those gaps, but the failure of the Scan Line Corrector in 2003 made its use too challenging in the context of this study.

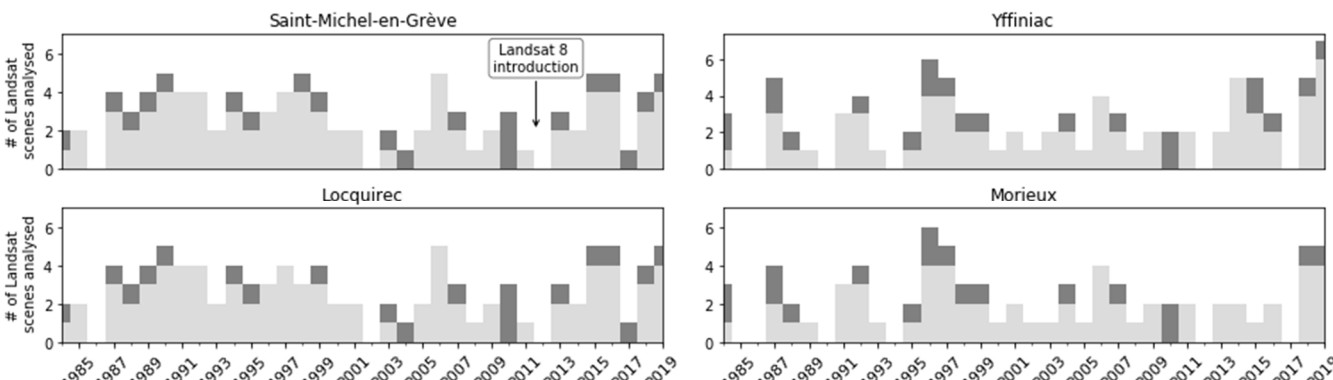

**Figure 2.** Number of images selected per year. Light grey indicates the images found during the summer months (from May to September included), whereas dark grey indicates the additional images found for the months of April and October.

### 2.3. Green Macroalgae Detection and Surface Estimates

For each selected scene, the Normalized Difference Vegetation Index (NDVI) (1) was computed.

$$\text{NDVI} = \frac{\text{NIR} - \text{Red}}{\text{NIR} + \text{Red}} \tag{1}$$

The NDVI is the most frequently used index to detect vegetation in remote sensing [33] and has been essential in analyzing temporal changes in vegetation, including the detection of macroalgae proliferation [22,34–36]. The index is based on the fact that

actively photosynthesizing vegetation, including algae, absorb light in the red wavelengths of sunlight, and reflect in the near-infrared (NIR). NDVI values range from −1 to 1 and are without units. High values approaching 1 mean a larger difference between the red and near infrared radiation recorded by the sensor, a condition associated with healthy and/or highly photosynthetically active vegetation. Low NDVI values mean that there is little difference between the red and NIR signals. Over vegetated areas, this indicates poor photosynthetic activity or low concentrations of plants. The ratio between the NIR and red bands can also compensate for changing light conditions and viewing angle.

In our study, we use NDVI to distinguish macroalgae stranded on the shores from sand, water and other elements on the beach, such as rocks or pebbles. Other remotely sensed indices, such as the Floating Algae Index (FAI), are also frequently used in the literature to detect algae, but their primary scope is the detection of algae at the water surface [23,24,37,38]. This was out of the scope of this analysis. Through visual inspection of the retained Landsat scenes, a threshold of a value of 0.22 was defined to mask pixels corresponding to macroalgae and to distinguish those from sandy areas and water.

Generating 35 years of NDVI time series required the use of two sensors: Landsat 5 TM and Landsat 8 OLI. The band ranges of the two sensors are similar, but the NIR band is considerably narrower for Landsat 8 (Table 1), to avoid atmospheric absorption effects. These differences mean that even if both sensors were looking at the same region of the electromagnetic spectrum, they could report different values in radiance. This would ultimately affect multi-temporal analysis. Cross-calibration between sensors was then required to allow multitemporal comparison. A scaling factor of 1.037 for producing a Landsat 5 equivalent to Landsat 7 NDVI products was used [39]. For adjusting Landsat 8 to Landsat 7 equivalent, the multiplication factor of 1.086 was used [40].

**Table 1.** Landsat 5 and 8 wavelengths range for red and infrared bands.

| | Landsat 5 TM | | Landsat 8 OLI | |
|---|---|---|---|---|
| **Descriptor** | **Band Number** | **Wavelengths [nm]** | **Band Number** | **Wavelength [nm]** |
| Red | Band 3 | 630–690 | Band 4 | 640–670 |
| Infrared | Band 4 | 770–900 | Band 5 | 850–880 |

Visual inspection of the NDVI products showed that the macroalgae detection needed to be improved, due to the detection of false positive at the sea surface. A mask using the Normalized Difference Water Index (NDWI) (2) was used in that scope.

$$\text{NDWI} = \frac{\text{Green} - \text{NIR}}{\text{Green} + \text{NIR}} \tag{2}$$

A threshold (−0.20) was defined by visual inspection to optimize the detection of green macroalgae, excluding false positives at sea, and including pixels in the sublittoral zone. A product combining the two masks was generated, highlighting vegetated areas on the foreshore. A basic formula enabled to retrieve the surface covered by those pixels, at different dates through the Google Earth Engine (GEE) platform. A .csv file with all the surface estimates by date was downloaded and then used for monthly, annual, and 5-year averages.

### 2.4. Additional Data

CEVA's estimates of *Ulva* macroalgae coverage were used for comparison with the Landsat-based surface estimates. CEVA uses aerial photography to quantify the algae surface. The aerial surveys are conducted in low tide conditions over a seven-month period (from April to October) for 30 priority sites in Brittany. CEVA does not publicly detail its methodology for surface estimations, but reports suggest that the algae surfaces are manually digitized through visual inspection. Yearly averages were computed for comparison with the Landsat-based yearly estimates.

Due to the relative coarse spatial resolution of Landsat imagery (30 × 30 m), we visually compared a selection of satellite scenes with high resolution aerial photography (0.71 cm × 0.71 cm). These aerial photography scenes were retrieved freely from the Institut National de Géographie (French National Institute of Geography). Over the 35 years studied, only one exact match in dates was found, on 7 June 2005 at Saint-Michel-en-Grève, and is shown for visually checking the robustness of the algae mask. At other dates, a few days separated Landsat and aerial photography scenes.

Time series data on nitrogen mean concentrations were also included in our study, given that nitrogen is the limiting factor for algae growth in Northern Brittany. Nitrogen mean concentrations were extracted for four water stations in Brittany. These water stations are located along the rivers draining the studied sites: the Roscoat (for Saint-Michel-en-Grève), the Douron (for Locquirec), Gouessant (for Morieux), and the Urne (for Yffiniac). The data was downloaded from the Environmental Observatory of Brittany (Observatoire de l'Environnement en Bretagne) website.

## 3. Results

### 3.1. Correspondence between Green Macroalgae Proliferation Detected by Aerial Photography and Landsat Imagery

The visual examination of aerial photography and Landsat 5 imagery on the 7 June 2005 (Figure 3) shows a very similar repartition of the macroalgae coverage on the beach of Saint-Michel-en-Grève. The green tide on that date is located in the eastern part of the foreshore. Dense accumulations are also visible in the fringing area (sublittoral zone) and on the upper part of the beach. The algae mask applied to the Landsat 5 scene clearly picks us pixels containing macroalgae content, indicating the NDVI is a good proxy for algae abundance.

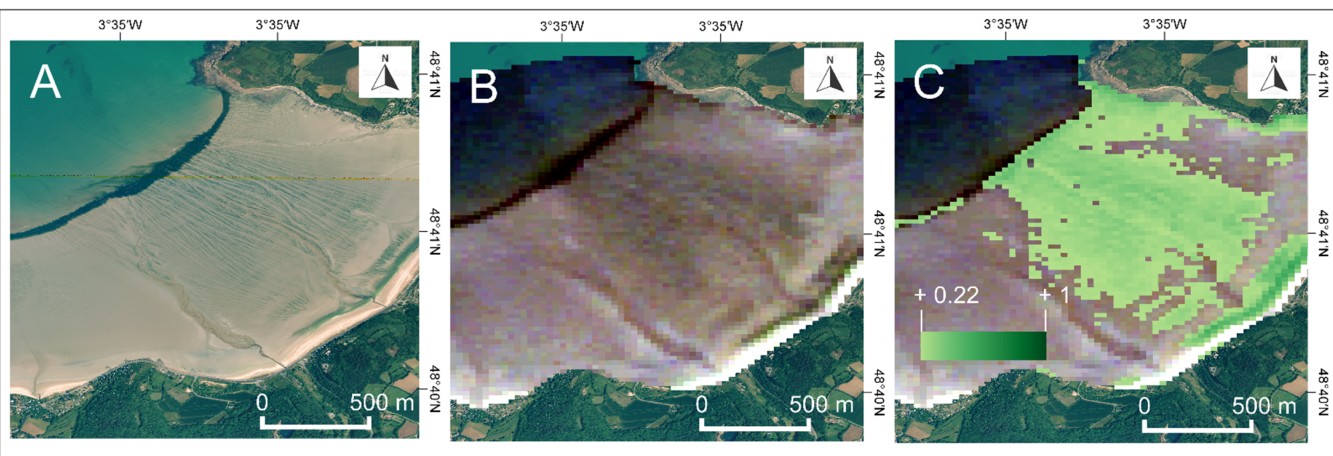

**Figure 3.** (**A**). High resolution (0.71 × 0.71 cm) aerial photography, at Saint-Michel-en-Grève. (**B**). Landsat 5 scene (30 × 30 cm) imagery, True Color Composite. (**C**). Algae mask with Normalized Difference Vegetation Index (NDVI) values from Landsat imagery. Both high resolution and Landsat scenes are from 7 June 2005. The aerial photography is freely provided the Institut National de Géographie (French National Institute of Geography).

### 3.2. Spectral Signature of Green Macroalgae

Algae show a reflectance peak in the green (~650 nm) and NIR wavelengths of light (~850 nm), and absorption in the red (~660 nm) (Figure 4A). Compared to healthy vegetation, however, the reflectance peak of green macroalgae in the NIR is lower. Macroalgae stranded on beaches are not highly photosynthetically active anymore, given that they are decomposing. The presence of water content could also result in the attenuation of the average reflectance of algae in the NIR. Further, unlike vegetation on land which has canopy and dense coverage, Landsat pixels containing stranded seaweed on the beach also contain the underlying substrate (Figure 3A). As shown in Figure 4, beach sand is not

highly reflective in the green and red infrared wavelengths of light; thus, mixed pixels of algae at different degrees of decomposition levels will show attenuated signal. As expected, beach sand shows a more even surface reflectance across Landsat central wavelength' bands, with also an increase of reflectance in the NIR, and then a decrease in the short-wave infrared bands. However, reflectance intensity in the NIR is lower on average than for algae. Water, on the other hand, is efficient for absorbing light in the near-infrared and, in this instance, reflects mostly in the green spectrum of light.

The spectral signatures plotted for pixels corresponding to algae in the sublittoral area (Figure 3B) show that the reflectance peak in the NIR depends on how much these pixels are filled with algae. On the sublittoral areas, algae are mixed with water, whereas, on the foreshore, they can be mixed with sand, suggesting less dense macroalgae patches. Considering that both water and sand reflect less in the NIR, thus, mixed pixels show lower reflectance peaks in the NIR.

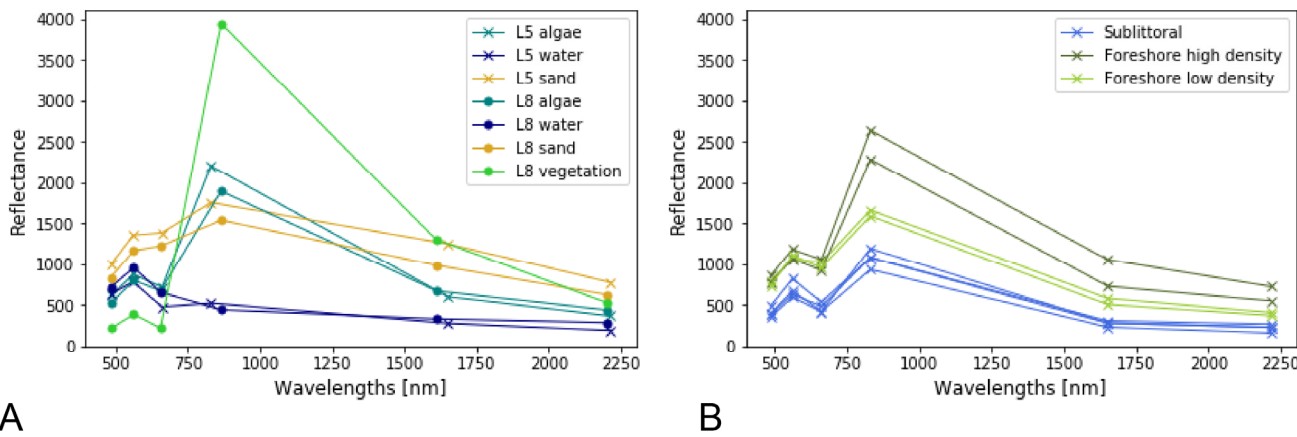

**Figure 4.** (**A**). Spectral signatures derived from the mean spectra of green macroalgae, sand, and water from monitored sites, at different dates. Spectral signature of healthy vegetation are shown as a reference and were retrieved nearby the studied sites. The -x- marker shows the central wavelengths of Landsat 8 (bands 2-3-4-5-6-7) and the -o- marker the central wavelengths of Landsat 5 (bands 1-2-3-4-5-7). The y-axis shows remote sensing surface reflectance (dimensionless). (**B**) Spectral signatures from macroalgae at different spots of Saint-Brieuc sites from Landsat 5 imagery from 14 June 1996. NDVI values for accumulations with high density of algae on the foreshore are typically above 0.40 and below 0.30 for areas with low density of algae.

### 3.3. Time Series Analysis

The macroalgae annual mean surfaces are detailed in Figure 5A for all four studied sites. Four years in total—1986, 2010, 2012, and 2017—have no record for any of our sites due to the absence of available imagery. The highest annual mean coverage (269 hectares) was reached in 1995 for the beach of Morieux. This corresponds to approximately 30% of the total beach area. The adjacent site of Yffiniac registered its maximum coverage in green macroalgae in 2005, with 215 hectares of estimated surface (17% of the beach area). The year 1985 registered the largest green tide average for both Saint-Michel-en-Grève (256 hectares) and Locquirec (30 hectares), covering, respectively, 60% and 21% of the beach areas. Overall, all sites show considerable year-to-year fluctuation in green tide surfaces. The variability for Saint-Michel-en-Grève is the highest, with a 30-fold range between minimum and maximum values. At Locquirec, green tides reach a twelve-time fold variation over our period, and eight-time fold for the beaches of the Saint-Brieuc bay. This year-to-year fluctuations are also a reflection of important inter-annual variability and skewed value distribution (Figure 5B).

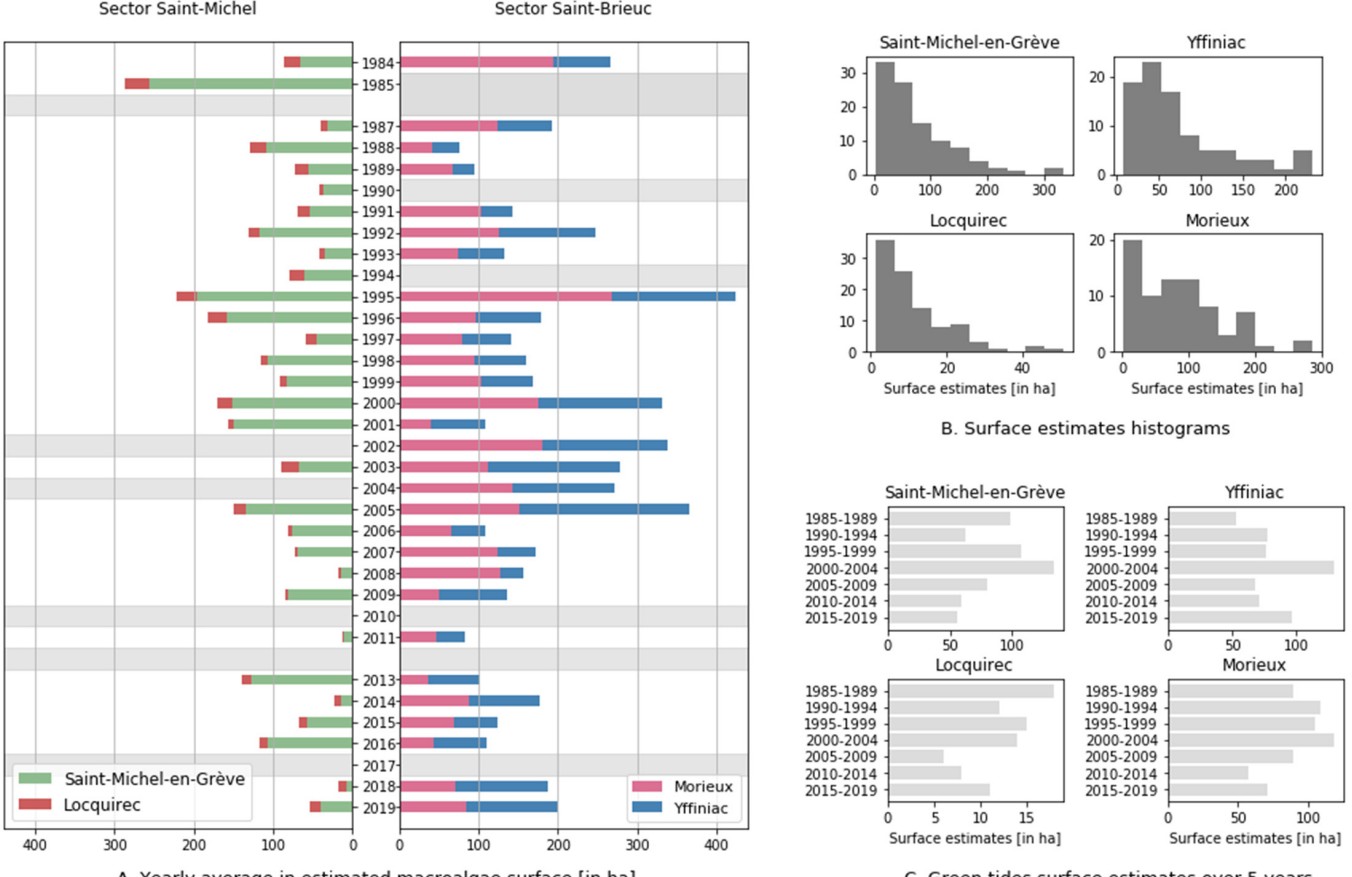

**Figure 5.** (**A**). Temporal variability of green tides for four sites in Northern Brittany: (**A**) annual mean averages, (**B**) frequency of surface estimates, and (**C**) surface estimates averages over 5-year periods.

Results of the Spearman correlation indicated that there was no significant trend in the yearly average of macroalgae estimated surfaces (rs = −0.31, *p* = 0.108, rs = 0.15, *p* = 0.455 and rs = −0.16, and *p* = 0.425, respectively, for Saint-Michel-en-Grève, Yffiniac, and Morieux) for all sites except Locquirec. At this site, a decreasing trend in *Ulva* surfaces is noticeable (rs = −0.50 and *p* = 0.0067).

Averaging the surface estimates over 5-year periods shows the longer-term dynamics of macroalgae surface (Figure 5C). The period from 2000 to 2004 registers a clear increase in terms of macroalgae coverage for all sites, except for Locquirec. Diverging trends can sometimes be observed between the two main sector studied: the early 1990s see an increase in the Bay of Saint-Brieuc (Yffiniac and Morieux sites) in mean surfaces, whereas the opposite trend is noticeable for the sites in Bay of Lannion. Conversely, the second half of the 1990s correspond to an increase in green tide coverage for Saint-Michel-en-Grève and Locquirec, whereas a very slight decrease is observable at both Morieux and Yffiniac beaches. The period from 2005 to 2009 marks a considerable decrease in green tide surfaces for all sites. This tendency is confirmed for the first half of the 2010s, except for Locquirec and Yffiniac, which register an increase in macroalgae coverage of, respectively, +31% and +5%. For the 2015–2019 period, an increase in green tide coverage is noticeable when compared to the precedent period for three sites out of four. Only Saint-Michel-en-Grève registers a slight decrease in green tide coverage (−5%) for the second half of the 2010s.

Monthly variations in green tide surfaces can be identified (Figure 6). At the end of the spring, the macroalgae accumulation starts in the sublittoral zone and as small patches on the foreshores. Peaks in macroalgae coverage usually occur between the end of May and mid-June, with a few exceptions. The month of June registered the highest mean surface

for all sites, apart for Locquirec. Mean surface estimates decrease from July onward, but this decrease is not linear for any of the sites.

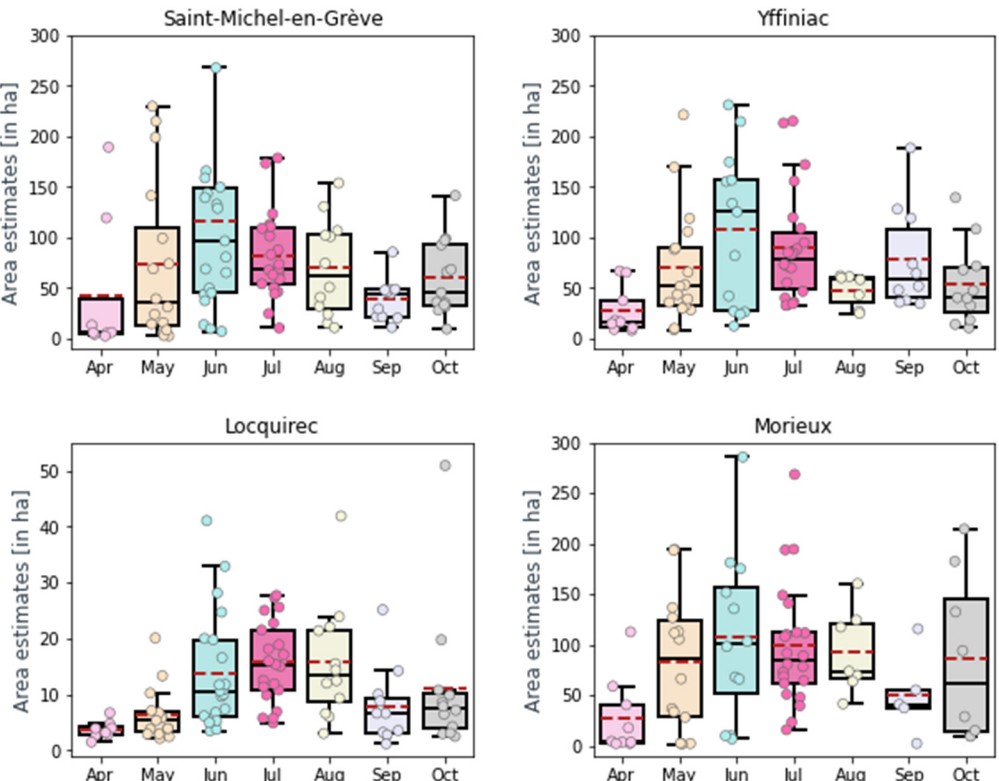

**Figure 6.** Seasonality of green tides, from April to October. The dashed bordeaux line indicates the mean value for each month, and the black one the median values.

### 3.4. Spatial Variability of Green Tides on the Shores

The areas where the *Ulva* macroalgae tend to accumulate on the beaches vary considerably. For Saint-Michel-en-Grève, the scene from 26 April 1984 shows an important accumulation on the upper beach slope which is less noticeable for the other images (Figure 7). In the case of Saint-Michel-en-Grève, dendritic patterns are clearly visible on the foreshore. Several images also show that apart from the foreshore itself, dense suspension of algae can be noted in the sublittoral zone, which corresponds to the wave breaking area. This is particularly noticeable for the imagery of 28 June 1995 and 14 June 1996. Different density rates in coverage are noticeable, with high NDVI values corresponding to dense patches of macroalgae.

Our analysis also shows important contrasts between sites in terms of peak years in green tides, despite them all being within 75 km of one another. For instance, little algae covered the beaches of the Saint-Michel sector in 2014 and 2018, whereas, in the case of Saint-Brieuc, these two years were marked by notably high concentrations.

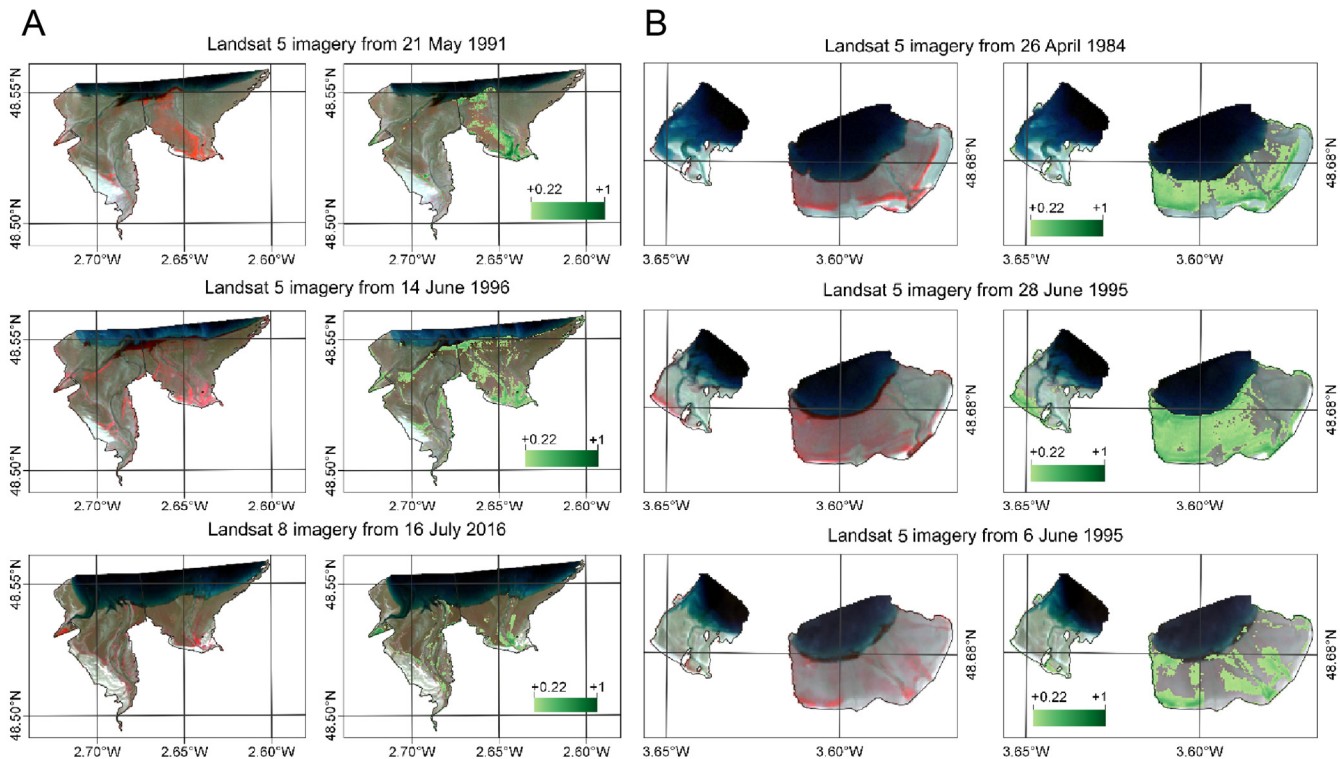

**Figure 7.** Examples of Landsat scenes at different dates showing green tides event. (**A**). Saint-Brieuc sites. (**B**). Saint-Michel sector. Left images show a Near-Infrared-Red-Green band combination, with vegetation showing in red pixels. Right images show the same combination overlaid with NDVI (threshold above +0.22), thus highlighting areas that were identified as macroalgae accumulation.

### 3.5. Comparison between Earth Observation and Aerial Photography Estimates

Comparing the recent trends, from 2002 to 2019, of annual mean surfaces derived from Landsat with CEVA's data shows good agreement levels in terms of trends, order of magnitude of the areal estimates and general variability in the data, as shown by Figure 8. For Yffiniac and Morieux, significant and positive correlations were found between the two time series (respectively: Spearman's r = 0.59, *p* = 0.016; Spearman's r = 0.69, *p* = 0.0029). Such a clear relation was not found for the sites of Locquirec nor Saint-Michel-en-Grève. However, at Saint-Michel-en-Grève, the data shows good agreement levels, especially from 2009 onwards. At Locquirec, where discrepancies between the two datasets are more notable, green tide surface estimates remain within the same range, except for three years: 2003, 2005, and 2009.

For Yffiniac and Morieux, CEVA's estimates confirm our analysis of an overall increase in *Ulva* green tide surfaces over the last few years. For Locquirec and Saint-Michel-en-Grève, however, no clear and steady dynamic is noticeable from 2002 onward.

The few remarkable discrepancies between our analysis and CEVA's estimates—for instance, in 2010 in Locquirec, 2019 in Morieux, and 2017 in Saint-Michel-en-Grève—can be attributed to the different methodologies used in determining the green macroalgae surface, coupled with the limited available EO imagery.

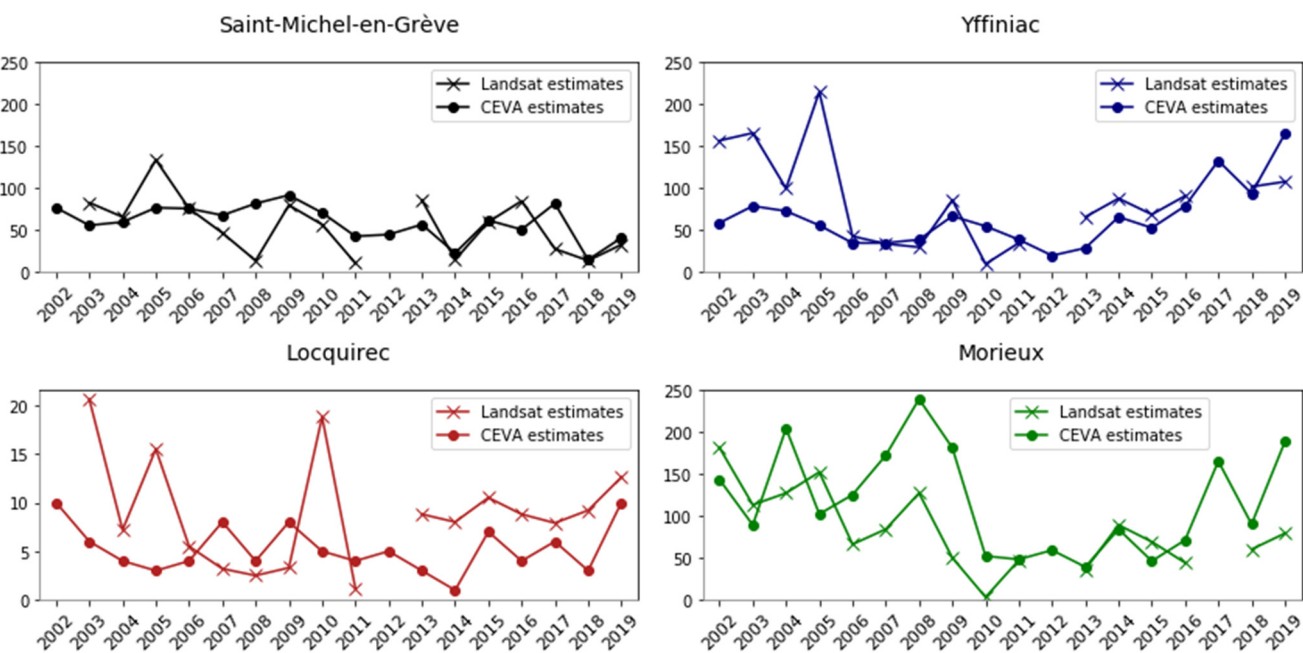

**Figure 8.** Comparison of green tide surface estimated by French Algae Technology and Innovation Center (CEVA) (through aerial photography) and derived from Landsat, for the four studied sites, from 2002 to 2019.

### 3.6. Nitrogen Mean Concentrations

Figure 9 shows the evolution of nitrogen mean concentration for four of the water stations impacting the study sites. Although concentrations decreased by 26–36% between 1995 and 2018, the 2018 values remained as high as 22–26 mg/L.

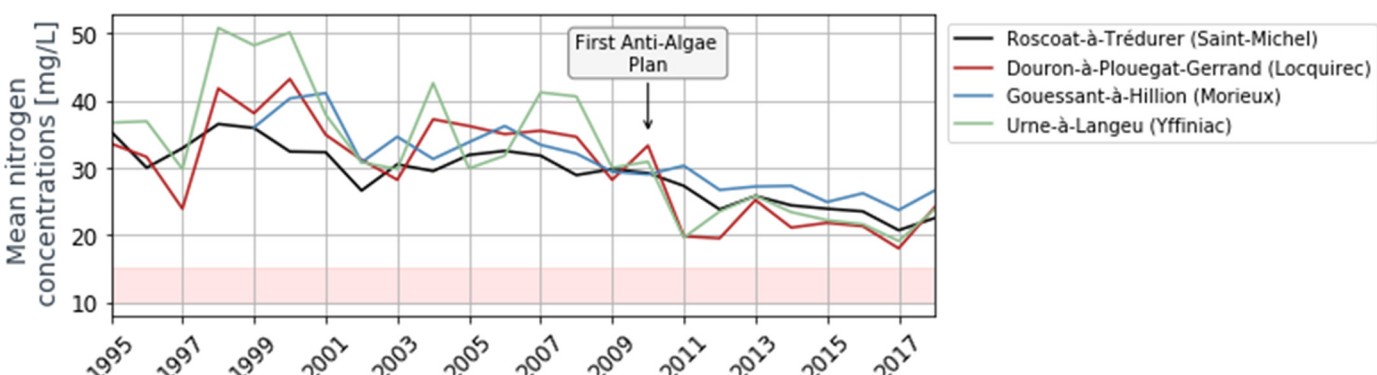

**Figure 9.** Nitrogen mean concentrations at four water stations near the study sites. These water stations were chosen for the availability of long and continuous time series. In red, the threshold values of 10–15 mg/L usually retained for a decrease in green tide surfaces. Source: Observatoire de l'Environnement en Bretagne, 2019.

## 4. Discussion

### 4.1. Reconstructing Green Macroalgae Long-Term Time Series

This study retrospectively estimated green macroalgae extents almost continuously over a 35-year period. Similar time-series analysis are not possible from in-situ measurements nor aerial photography due to the lack of long-term historical records. In Northern Brittany, the four studied sites have one of the most systematically and longest collected monitoring program, based on aerial photography of green tide coverage. Nonetheless, these time series are only available from 2002 onwards and the methodology for algae detection is not detailed. For these reasons, long-term EO mission, such as Landsat, are valuable tools even in well-monitored areas to show the patterns and dynamics of macroal-

gae blooms over 35 years. Nevertheless, the collection of ground truth data remains a very valuable source to assess the accuracy of EO data and to refine the analysis of macroalgae on the foreshore by providing information on algae species and biomass density.

Our findings show that large *Ulva* stranding events happened throughout the 35-year period, with significant interannual and year-to-year variability in macroalgae surface estimates. Proliferation variability is due to the several factors that influence green macroalgae growth and are widely highlighted in the existing literature. Certain meteorological conditions during the winter and spring seasons (such as the occurrence of storms, tides, and high precipitation levels) and high levels of macroalgae stock over the fall are correlated with strong green tide years [10–12]. Monitoring efforts that focus on precise quantification of green tide surfaces and diurnal variability should also take into account inconsistencies in the temporal acquisition of Landsat scenes. Notably, orbital changes in the overpass time of Landsat 5 mission might contribute to differences in NDVI values over long time series [41], but no evidence was found for this in our specific case.

No significant trends—either increases, stabilization or decreases—in macroalgae coverage are noticeable for the three largest study sites throughout the entire period (Saint-Michel-en-Grève, Yffiniac and Morieux). Only at Locquirec, a decreasing trend in green tide surfaces is registered, although this trend is not steady over the period. As a rule, years of low algae proliferation (1996–1997–1998–1999 for the beach of Morieux; Figure 4) can be interspersed with years of strong tides (1995 and 2000). This is consistent with existing studies, notably CEVA's reports [31,42], the audit of the first Anti-Algae Plan [32], and IFREMER (the French National Institute for Ocean Science)'s publications [11]. Nonetheless, it appears that the very large green tides estimated for the period between 1984 and 2005 have not been registered as often after 2005.

Moreover, the direct comparison of green tide surfaces between Landsat and CEVA datasets (for the 2002–2019 period) shows a good fit between satellite remotely obtained data and high-resolution aerial photography. The visual comparison between a high-resolution imagery and Landsat scene also illustrates the robustness of estimating green tide abundance with Landsat imagery. Despite coarser spatial resolution, the algae mask based on NDVI and NDWI is able to detect areas with green macroalgae content on the beach.

### 4.2. Decrease in Nitrogen Concentrations and Influence on Green Tides Surfaces

Despite a registered decrease of nitrogen values in the rivers draining the study sites (Figure 8), this did not materialize in a drop in green tide surfaces. This could be explained by the fact that the decrease in nitrogen is not substantial enough to influence the algae surface and biomass because the relationship between nitrogen and algae's bloom is non-linear [11]. Indeed, for all but two of the water stations draining our study's beaches, mean concentrations of nitrogen were above 20 mg/L in 2018. Indeed, for several stations that year, mean values approximated 30–35 mg/L (Observatoire de l'Environnement en Bretagne, 2019). According to CEVA's and IFREMER's studies, only nitrogen concentrations below 10–15 mg/L for a normal hydrological year would translate into a decrease in green tide surface and biomass [43,44]. Mathematical modellings on the mean nitrogen concentration and macroalgae biomass have established the required target values in nitrogen for a reduction of macroalgae surfaces [11,45]. In the case of the Douron river, the main waterway draining the Locquirec beach and responsible for its green tides, only nitrogen concentration levels below that 20 mg/L would result in a decrease in macroalgae surfaces [11].

Hence, the interannual fluctuations in green tide surfaces should be attributed mainly to climatological factors at this stage, rather than to a decrease in nitrogen concentrations in the rivers draining the studied beaches. Efforts in reducing green tides have, thus, not shown any major improvements for the studied sites yet.

*4.3. Uncertainties and Next Steps*

Estimating green macroalgae surface through Earth Observation relies greatly on the availability of images in low tides conditions, and is hindered by cloud coverage. Given the low revisit time of both Landsat 5 and 8 satellites—16 days—and the frequent presence of clouds over the coasts of Northern Brittany, satellite data for some of the years considered in the study were not available. Thus, macroalgae surfaces could not be estimated. For many other years, only one or two scenes were available over an entire season. For these reasons, the annual mean surface estimates should be carefully interpreted.

Sentinel-2A (since 2015) and Sentinel-2B (since 2017) have a combined revisit time of 2–5 days, allowing for a higher number of images to be made available over the areas of interest. The higher spatial resolution—10 m for the blue, green, red, and infrared bands; and 20 m for other visible and near-infrared bands—could also play a role in improving surface estimates. If Sentinel-2 data were to be combined with Landsat 8, this would result in even higher number of available scenes. However, the cloud coverage remains a hinderance when working with optical remote sensing data. Since 2014, the Copernicus program has launched Sentinel-1, a mission acquiring radar imaging with a revisit time of 12 days. Radar microwave data has the capacity to acquire imagery day and night and does not heavily depend on weather conditions. The use of Sentinel-1 imagery in monitoring green algae stranded on the foreshores could also be a factor for improving green algae surface estimates [46]. Historic algae blooms, such as the *Sargassum* stranding events reported since 2011 in the Caribbean region, could be further investigated to test the applicability of our detection method for these accumulation events.

## 5. Conclusions

We present a new, simple method to detect and monitor green macroalgae blooms using freely available Landsat imagery and GEE. Due to their distinct spectral characteristics, the stranded macroalgae can be detected through EO. A first application to four beaches in Northern Brittany revealed the temporal and spatial dynamics of algae blooms between 1984 and 2019.

We found that despite an observed decrease in nitrogen concentration levels, *Ulva* blooms were still going strong. This highlights the importance of continuous and long-term monitoring of macroalgae surface estimates, possibly with the addition of EO data collected by Landsat and Sentinel satellites. It also underscores the need for more ambitious targets in nitrogen concentration reduction.

Additionally, our method could be transferred to other systems of interest. Future research may include the study of other historic events, global assessments, and even early-detection systems using a combination of Landsat and Sentinel imagery.

**Author Contributions:** Conceptualization: L.S. and Y.-F.L.L.; methodology, analysis and writing–original draft preparation: L.S.; supervision and writing–review and editing: T.v.E.; writing–review and editing: L.B. and Y.-F.L.L. All of the authors thoroughly reviewed and edited this paper. All authors have read and agreed to the published version of the manuscript.

**Funding:** This research received no external funding.

**Institutional Review Board Statement:** Not applicable.

**Informed Consent Statement:** Not applicable.

**Data Availability Statement:** Data generated and used during the study has been uploaded on the 4TU.ResearchData repository and is available at https://data.4tu.nl/articles/dataset/Dataset_on_green_tide_surfaces_over_Brittany_using_Landsat_imagery_for_35_years_of_monitoring/13747231/2 (accessed on 1 March 2021). Sample codes used for the analysis are available at https://github.com/louiseschreyers/Green_tides_Brittany.git (accessed on 8 February 2021).

**Acknowledgments:** The data provided by the U.S. Geological Survey, CEVA, IGN, and Observatoire de l'Environmental en Bretagne is greatly appreciated. We thank the reviewers for their constructive feedback which helped improved the manuscript. We thank Louise Sampagnay and Nicolas Prudhomme for proofreading the manuscript.

**Conflicts of Interest:** The authors declare no conflict of interest.

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
