# Peer review of "Spotting Green Tides over Brittany from Space: Three Decades of Monitoring with Landsat Imagery"

_remotesensing, doi:10.3390/rs13081408_

Round 1

Reviewer 1 Report

In this paper, the authors use GEE to study the spatial and temporal changes of green tides at four sites in Northern Brittany, France. The results show the interannual and seasonal fluctuations of the algae distribution, and indicate that there is no decrease in macroalgae surface on three out of four studied sites. I think this study can provide useful information for the government-decision-making. However, some issues need to be considered: 1, the authors should provide more descriptions about the macroalgae tides appeared in the study areas? What type of the macroalgae? When is the growing season? The frequency? The spectral difference between the macroalgae and other plants? 2, quantitative evaluation of their mapping results are necessary. For example, the authors should calculate the classification accuracies.

Reviewer 2 Report

The manuscript entitled “Spotting green tides over Brittany from space: three decades of monitoring with Landsat imagery” is about the illustration of a new, simple method to detect and monitor algae blooms using freely available Landsat imagery and Google Earth Engine. This manuscript is well-written. It has a clear objective. The associated findings do have important application values. Therefore, I would recommend the manuscript to be accepted for publication, provided that the following two main issues could be addressed satisfactorily.

First, in the “materials and methods” section, please state whether there are any robustness checks of the findings. In the “results” section, please present the robustness check results. Readers may need this information to assess whether the proposed method is reliable or not. I realize that the authors have tried to do it in section 3.4 “comparison between Earth Observation and aerial photography estimates.” But, this is not enough.

Second, in the “discussion” section, please expand the content. It is necessary to illustrate in what ways the proposed methods are better than the other existing methods in tracing algae blooms. Besides, please compare the findings with those in other studies. This information is important to persuade people in trying the method proposed in this study.

Reviewer 3 Report

The article “Spotting green tides over Brittany from space: three decades of monitoring with Landsat imagery” have applied remote sensing in analyzing the algal blooms. However, there are many concerns that hesitate me to accept this paper. The introduction is poor. It does not provide enough evidence to convince the necessity of research and is jumping from different irrelevant topics. For example: “The largest green tide observed to date occurred along the Yellow Sea coast during the Olympic Games sailing competition of 2008, in the Chinese region of Qingdao” doesn’t connect. There are many issues that do not relate to the proposed objective of the research.

There are many grammatical issues. For example: in Abstract

Line 21:” in the mean algae estimates on” needs to be rephrased

Line 22: “does not correlate to reduction in Earth Observation” needs to be rephrased

Introduction

Line 33: “Sargassum blooms” it has been for the first time introduced with no background about the Sargassum species.

In many places, authors miss the proper references. For example line 54:” in 2010 by the French government.”

The results are mixed with dissection, then again there is a short discussion headline.

Round 2

Reviewer 2 Report

The authors have addressed all of my previous comments and suggestions. Therefore, I recommend this manuscript be accepted for publication.

Author Response

Thank you very much for the positive feedback and all the valuable comments during the reviewing process.

Reviewer 3 Report

I believe the authors have revised the paper to best of their ability. The new version shows a significant improvement. It may be accepted for publication.

Author Response

(The authors gave the same response as above.)
